**Subject Category:**
Biology (whole organism)

physiology/biometrics/developmental biology

barley, deer, nutrition, puberty, spermatogenesis, spermatozoa

**Authors for correspondence:**
José Luis Ros-Santaella
e-mail: rossantaella@gmail.com
Eliana Pintus
e-mail: eliana.pintus27@gmail.com

# High-energy diet enhances spermatogenic function and increases sperm midpiece length in fallow deer (Dama dama) yearlings

José Luis Ros-Santaella[1], Radim Kotrba[2,3]
and Eliana Pintus[1]

[1]Department of Veterinary Sciences, Faculty of Agrobiology, Food and Natural Resources, and [2]Department of Animal Science and Food Processing, Faculty of Tropical AgriSciences, Czech University of Life Sciences Prague, Kamýcká 129, 16500 Prague 6-Suchdol, Czech Republic
[3]Department of Ethology, Institute of Animal Science, Přátelství 815, 10400 Prague 10-Uhříněves, Czech Republic

JLR-S, 0000-0002-4186-7730; EP, 0000-0002-8496-0820

Nutrition is a major factor involved in the sexual development of livestock ruminants. In the male, a high-energy diet enhances the reproductive function, but its effects on the underlying processes such as spermatogenic efficiency are not yet defined. Moreover, the possible changes in sperm size due to a supplemented diet remain poorly investigated. The main goal of this study was to evaluate whether a high-energy diet affects the spermatogenic activity, epididymal sperm parameters (concentration, morphology, morphometry and acrosome integrity) and blood testosterone levels in fallow deer yearlings. For this purpose, 32 fallow deer were allocated into two groups according to their diet: control (pasture) and experimental (pasture and barley grain) groups. Fallow deer from the experimental group showed a significant increase in the Sertoli cell function and sperm midpiece length, together with a higher testicular mass, sperm concentration and percentage of normal spermatozoa than the control group ($p < 0.05$). We also found a tendency for higher blood testosterone levels in the animals fed with barley grain ($p = 0.116$). The better sperm quality found in the experimental group may be related to their higher efficiency of Sertoli cells and to an earlier onset of puberty. The results of the present work elucidate the mechanisms by which dietary supplementation enhances the male sexual development and might be useful for better practices of livestock management in seasonal breeders.

# 1. Introduction

Nutrition is one of the main factors involved in the sexual development and reproductive functions of livestock [1]. In the adult male, for instance, high-energy diets increase the libido, testis size, seminiferous tubule diameter and sperm production [2–4]. Likewise, in the young male, high nutritional intake promotes an earlier onset of puberty, with relevant economic implications for the livestock industry [5,6]. Several studies indicate that enhanced early-life nutrition increases the body mass, testes mass, testosterone levels and epididymal sperm reserves [7–11], because high-energy diets may hasten the onset of the spermatogenesis. In this context, the use of bulls at an earlier age reduces the production costs, shortens the generation interval and may increase the genetic gains [9]. Nevertheless, the effects of nutritional supplementation on the spermatogenic efficiency as well as on the size of the sperm structures in non-sexually mature males are still little known.

The fallow deer (*Dama dama*) is a seasonal breeder (breeding season from the end of September till the beginning of November; [12]), which has been exploited in the last decades as a farm animal mainly for its venison attributes [13]. According to the gonadal mass and spermatogenic activity, males are classified into three age groups: fawns, yearlings and adults (more than 2 years of age) [14]. Male fallow deer reach puberty at about 16 months of age and, from a physiological point of view, they can achieve fertile mating [12,14]. However, yearlings most probably do not breed at this age because the social hierarchy in the herd may prevent mating [12]. In the yearlings, the maximum value of testes mass is reached in October followed by a gradual decrease till the minimum value in April (up to threefold decrease; [14]). Moreover, while adult fallow deer males drastically decrease their feeding activity before and during the breeding season, yearlings increase their food intake in response to an improvement in forage quality and availability [15]. For this reason, fallow deer yearlings represent a good model to investigate the effects of a high-energy diet on the male reproductive traits in this species. Moreover, a dietary supplementation administered prior to the rutting season allows studying more in depth the effects of nutrition on spermatogenic activity, sperm traits and blood testosterone levels in seasonal breeders. To the best of knowledge, the studies addressing the relationship between nutrition and sexual development in fallow deer are scarce and limited to the female sex [16].

The aims of the present study were to evaluate the effects of high-energy diet on body mass, testis mass, spermatogenic activity, sperm traits and testosterone levels in non-sexually mature fallow deer. Previous studies in red deer highlight the relevance of spermatogenesis and sperm size on sperm fitness [17,18]. In order to increase the dietary intake, fallow deer diet was supplemented with barley grain. Barley ranks third as a readily degradable cereal for ruminants and it is irreplaceable by any other grain for producing capacious rumen microbial yields together with a high protein and amino acids content [19]. Moreover, it is worth noting that the nutritionally essential amino acids found in barley grain (i.e. lysine, methionine, cysteine and tryptophan; [19]) exert positive effects on spermatogenesis, sperm quality and fertility [20–22]. Taking into account that spermatogenesis in small domestic ruminants (i.e. ram and goat) lasts on average 48.5 days [23], the dietary supplementation was administered six months prior to the rutting season. The results from this work can reveal new insights into the importance of nutrition at puberty in seasonal breeders and its possible implications in the reproductive performance of adults.

# 2. Material and methods

## 2.1. Reagents

Reagents were purchased from Sigma-Aldrich (Prague, Czech Republic), unless otherwise stated.

## 2.2. Animals, treatments and diet

Thirty-two fallow deer were used in the present study and allocated in April (fallow deer age approx. 10 months) into two balanced groups according to their body mass. One group ($N = 16$) was fed ad libitum on pasture only (control diet), whereas the second group ($N = 16$) was fed ad libitum on pasture with whole barley grain supplementation (high-energy diet). All animals had ad libitum access to water and mineral supplementation (Premin®, VVS Verměřovice s.r.o., Czech Republic). The barley grain supplementation consists of 0.4 kg animal$^{-1}$ day$^{-1}$ at the beginning of the experiment (3 April), then increased to 0.6 kg animal$^{-1}$ day$^{-1}$ from 27 July till the end of experiment

**Table 1.** Feed composition of diets provided to fallow deer yearlings.

|  | pasture[a] | barley grain |
|---|---|---|
| dry matter (DM; g kg$^{-1}$ fresh matter) | 321.70 | 890.30 |
| crude protein | 127.39 | 112.67 |
| crude fat | 19.07 | 24.41 |
| crude fibre | 316.12 | 66.83 |
| nitrogen-free extract | 45.25 | 77.1 |
| acid detergent fibre | 352.30 | 72.58 |
| acid detergent lignin | 50.01 | 8.30 |
| neutral detergent fibre | 654.25 | 303.99 |
| ash | 84.90 | 25.10 |

[a]Mean value from three samples collected during the experiment. All values are expressed as g kg$^{-1}$ DM, except for nitrogen-free extract (% DM).

**Table 2.** Estimated energetic values of diets provided to fallow deer yearlings. DM, dry matter.

|  | pasture | barley grain |
|---|---|---|
| organic matter digestibility (%) | 66.49 | 81.82 |
| energy digestibility (%) | 63.56 | 79.47 |
| gross energy (MJ kg$^{-1}$ DM) | 18.14 | 18.47 |
| digestible energy (MJ kg$^{-1}$ DM) | 11.53 | 14.68 |
| metabolizable energy (MJ kg$^{-1}$ DM) | 9.34 | 11.89 |

(i.e. October). The average chemical composition of pasture and barley grain is described in table 1. During the experiment, pasture samples were collected three times from three locations within each paddock (at the beginning, middle and at the end of experiment, respectively). All diet samples were dried using a freeze-drying method (Freeze dryer ALPHA 1-4 LSC, Martin Christ Gefriertrocknungsanlagen GmbH, Germany) and the average nutrient composition was analysed as described by Jančík *et al*. [24]. The chemical composition of pasture and barley grain was determined according to the following methods: dry matter (DM, oven drying for 6 h at 105°C to a constant weight); crude protein (calculated as nitrogen × 6.25); crude fat (6 h extraction with petroleum–ether using Soxtec 1043, FOSS Tecator AB, Höganäs, Sweden); nitrogen (Kjeldahl method, Kjeltec AUTO 1030 Analyser, Höganäs, Sweden) according to AOAC 976.05 [25]; acid detergent fibre (ADF) and lignin (determined according to AOAC 973.18, [25]); neutral detergent fibre (NDF, analysed in the presence of sodium sulfite and with α-amylase, [26]); and ash (oven drying for 6 h at 550°C). The energetic values of pasture and barley grain were estimated as follows [27–29]: organic matter digestibility (OMd, %) of pasture = 90.8 − 0.091 × crude fibre (g kg$^{-1}$ DM) + 0.035 × crude protein (g kg$^{-1}$ DM) and OMd of barley = 92.2 − 0.149 × crude fibre (g kg$^{-1}$ DM) + 0.0074 × crude protein (g kg$^{-1}$ DM) − 0.050 × ash (g kg$^{-1}$ DM); energy digestibility (Ed, %) for pasture = −0.068 + 0.957 × OMd (%) and Ed for barley = −3.94 + OMd(%) + 0.0104 × crude protein (g kg$^{-1}$ DM) + 0.0149 × crude fat (g kg$^{-1}$ DM) + 0.0022 × NDF (g kg$^{-1}$ DM) − 0.0244 × ash (g kg$^{-1}$ DM); gross energy (GE, MJ kg$^{-1}$ DM) = 17.3 + 0.0617 × crude protein (% DM) + 0.2193 × crude fat (% DM) + 0.0387 × crude fibre (% DM) − 0.1867 × ash (% DM) + correction coefficient (correction coefficient 0 and 0.15 for pasture and barley, respectively); digestible energy (DE, MJ kg$^{-1}$ DM) = GE × Ed/100; metabolizable energy (ME, MJ kg$^{-1}$ DM) = DE × 0.81 (table 2). During the breeding season (October), fallow deer (approx. 16 months old) were slaughtered between 9.00 and 12.00. In order to minimize the time interval between the animal death and the sample collection, samples were processed following the same order of slaughtering. In this way, we guaranteed that all individuals were processed approximately after the same time interval after their death. Moreover, the researchers who performed the testicular and sperm analyses were blind about the group assignment of the animals.

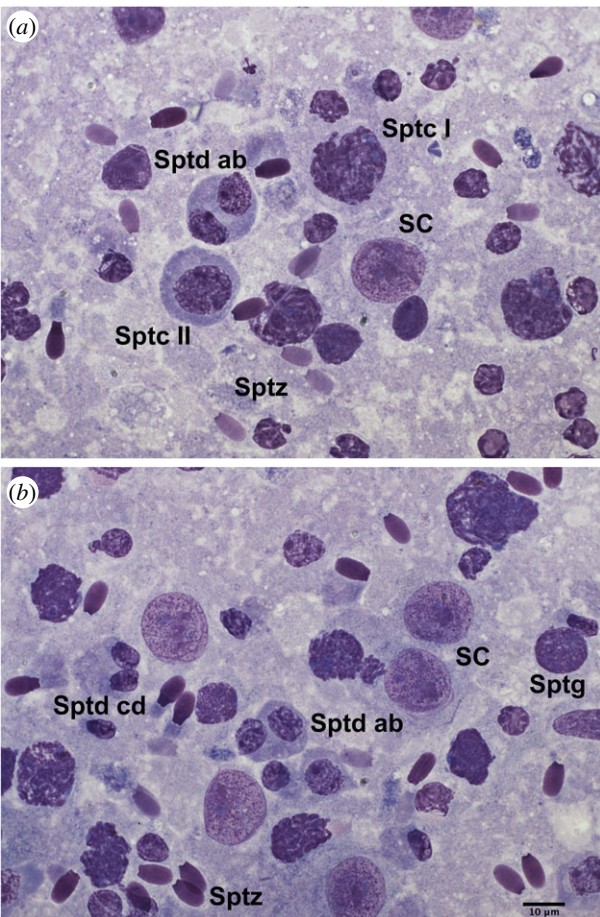

**Figure 1.** Testicular germ cells and Sertoli cells from fallow deer yearlings. (*a,b*) SC, Sertoli cell; Sptg, spermatogonium; Sptc I, primary spermatocyte; Sptc II, secondary spermatocyte; Sptd ab, round spermatid; Sptd cd, elongated spermatid; Sptz, spermatozoon.

## 2.3. Body mass, combined testes mass and gonadosomatic index

Body mass (measured at the beginning and at the end of experiment) was determined in handling facilities at farm by Tru-Test EziWeight tenzometric scale (Tru-Test Group, Auckland, New Zealand) with an accuracy of 0.1 kg. The testes (within the scrotum) were removed right after the slaughter and transported to the laboratory at 17°C. Testicular mass was recorded to the nearest 0.1 g using an electronic balance (EK-600G, LTD, Japan) after removing the epididymis and the spermatic cord with a surgical blade. The gonadosomatic index was calculated as: [(combined testes mass/body mass) × 100].

## 2.4. Testicular cytology

Cytological samples were collected from each testis using the fine needle aspiration cytology (FNAC) technique and stained with Hemacolor (Merck, Darmstadt, Germany) as previously described [30]. Quantitative assessment of spermatogenesis was performed on at least 200 spermatogenic and Sertoli cells per testis using bright-field microscopy (Nikon Eclipse E600, Nikon, Japan; 100× objective). The spermatogenic and Sertoli cells were classified based on their morphology as shown in figure 1. Then, the percentage of spermatogenic cell subtypes was determined and several testicular cytological indices were assessed as follows: (i) the Sertoli cell index (SEI), which is the percentage of Sertoli cells per total germ cells and estimates the spermatogenic activity; (ii) the spermatozoa–Sertoli index (SSEI), which is the number of spermatozoa per Sertoli cell; (iii) the meiotic index (MI), which is the ratio of round spermatids to primary spermatocytes and estimates the meiotic germ cell loss; (iv) the ratio of elongated spermatids to round spermatids (ES/RS), which estimates the post-meiotic germ cell loss; (v) the ratio of elongated spermatids to total germ cells (ES/GC), which estimates the overall germ cell loss; (vi) the ratio of round spermatids to Sertoli cells (RS/SC), (vii) the ratio of elongated

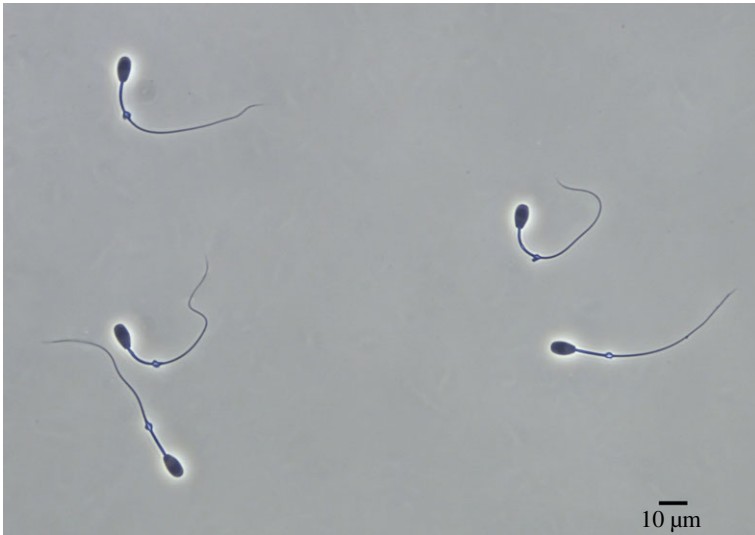

**Figure 2.** Fallow deer spermatozoa (normal morphology).

spermatids to Sertoli cells (ES/SC), which both estimate the Sertoli cell function; and (viii) the ratio of total germ cells to Sertoli cells (GC/SC), which estimates the Sertoli cell workload capacity [18,30].

## 2.5. Blood plasma testosterone levels

Blood samples were collected during slaughter procedure when animals were bled by cutting of the jugular vein and carotid arteries into ethylenediaminetetraacetic acid (EDTA) vials. After centrifugation at 3500$g$ for 20 min at 4°C, the supernatant was transferred into a new vial and stored at −80°C till analyses. Testosterone levels were determined by an enzyme immunoassay with a double-antibody technique as described by Roelants *et al.* [31]. The assessment of testosterone levels was carried out in duplicate and expressed as ng ml$^{-1}$.

## 2.6. Sperm analyses

Sperm samples were recovered from the epididymal caudae and suspended in 0.5 ml of phosphate-buffered saline solution. Sperm concentration was assessed using a Bürker chamber. A small aliquot of sperm sample was recovered from the cauda of both epididymides and fixed in 2% glutaraldehyde−0.165 M cacodylate/HCl buffer (pH 7.3) in order to evaluate the sperm morphology, morphometry and acrosome integrity under phase-contrast microscopy (Nikon Eclipse E600, Tokyo, Japan; 40× objective). Normal sperm morphology was calculated as the proportion of spermatozoa without any morphological abnormality (e.g. abnormal and detached head, bent midpiece, coiled tail, proximal cytoplasmic droplet, etc.). An example of normal sperm morphology in fallow deer is shown in figure 2. Both for sperm morphology and acrosome integrity, 200 spermatozoa per male were evaluated. Sperm morphometry was assessed as previously described [32]. Briefly, sperm pictures were taken with a digital camera (Digital Sight DSFi1, Nikon, Japan) under phase-contrast microscopy, and sperm were then measured using the ImageJ software (NIH, USA). The following sperm traits were determined: head length and width, head area, head perimeter, flagellum length, midpiece length, principal plus terminal piece length and total sperm length. Twenty-five spermatozoa were measured per male.

## 2.7. Statistical analyses

Statistical analyses were performed by SPSS 20.0 statistical program (IBM Inc., USA). The normal distribution of data was checked by the Shapiro−Wilk test. The paired sample $t$-test was used to check for differences within groups between the beginning and the end of the experiment. The independent sample $t$-test was used to check for differences between groups when data were normally distributed, otherwise the Mann−Whitney $U$-test was used. Data are shown as mean ± s.d. Statistical significance was set at $p < 0.05$.

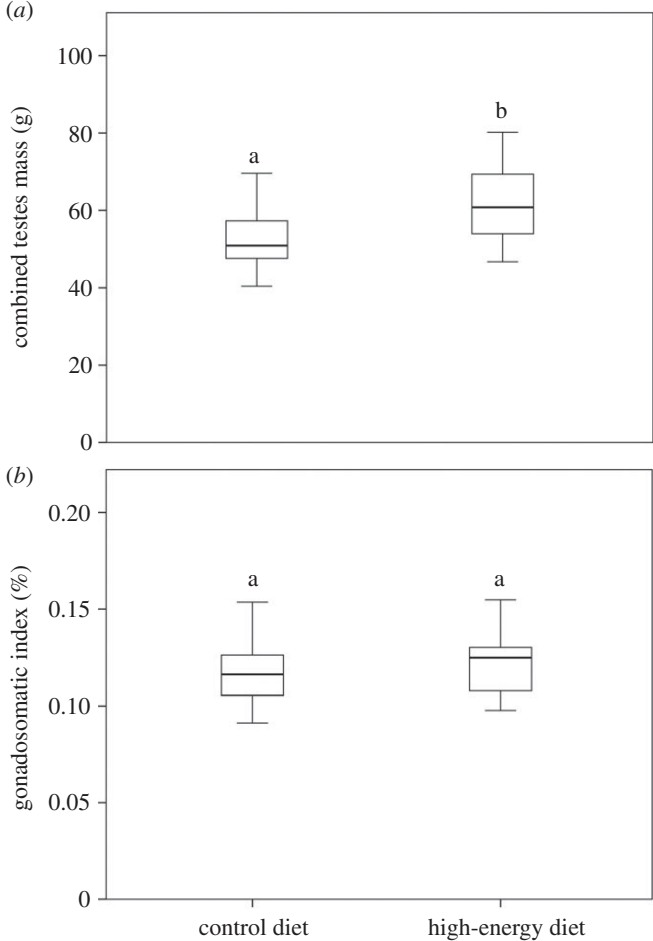

**Figure 3.** (a) Combined testes mass and (b) gonadosomatic index of fallow deer yearlings fed with pasture (control diet, $N = 15$) or pasture with barley grain supplementation (high-energy diet, $N = 16$). Different letters above the box plots indicate significant differences between groups ( $p < 0.05$).

## 3. Results

### 3.1. Body mass, combined testes mass and gonadosomatic index

At the beginning of the experiment, there was no difference in the body mass between treatments ($28.29 \pm 1.84$ kg versus $28.29 \pm 1.67$ kg, for control and high-energy diets, respectively; $p = 0.992$). Conversely, at the end of the experiment, fallow deer from the high-energy diet group showed higher body mass than those from the control group ($50.34 \pm 2.77$ kg versus $45.27 \pm 1.79$ kg, respectively; $p < 0.001$). Despite the fact that fallow deer from both treatments increased their body mass during the experiment ( $p < 0.001$), on average fallow deer from the high-energy diet group showed a significantly higher percentage of gained body mass than those from the control group ($78.13 \pm 8.11\%$ versus $60.45 \pm 8.79\%$, respectively; $p < 0.001$). Moreover, we found that combined testes mass was higher for animals from the high-energy diet group than those from the control group ( $p = 0.015$; figure 3a), but there were no differences in the gonadosomatic index between treatments ( $p = 0.380$; figure 3b).

### 3.2. Testicular cytology

All testicular samples showed normal and full spermatogenic activity. There was no difference in any spermatogenic cell subtype between treatments ( $p > 0.05$; table 3). By contrast, the SEI and the indices related to the functionality and workload capacity of the Sertoli cells (i.e. SSEI, RS/SC, ES/SC and GC/SC) differed significantly between treatments ( $p < 0.05$; table 3), being the indices related to the Sertoli cell functionality and workload capacities greater in the high-energy diet group. Moreover, all

**Table 3.** Percentages of germ cell subtypes and testicular indices of fallow deer yearlings fed with pasture (control diet, $N = 16$) or pasture with barley grain supplementation (high-energy diet, $N = 16$). Values are shown as mean $\pm$ s.d. SEI, Sertoli cell index; SSEI, spermatozoa–Sertoli index; MI, ratio of round spermatids to primary spermatocytes (meiotic germ cell loss); ES/RS, ratio of elongated spermatids to round spermatids (post-meiotic germ cell loss); ES/GC, ratio of elongated spermatids to total germ cells (overall germ cell loss); RS/SC, ratio of round spermatids to Sertoli cells (Sertoli cell functionality); ES/SC, ratio of elongated spermatids to Sertoli cells (Sertoli cell functionality); GC/SC, ratio of total germ cells to Sertoli cells (Sertoli cell workload capacity). Italics indicate significant differences between groups ($p < 0.05$).

| assessed parameters | control diet | high-energy diet | *p*-value |
|---|---|---|---|
| *spermatogenic cell subtypes* | | | |
| spermatogonia (%) | 1.82 $\pm$ 0.57 | 2.12 $\pm$ 0.47 | 0.119 |
| primary spermatocytes (%) | 18.03 $\pm$ 3.59 | 18.21 $\pm$ 3.21 | 0.876 |
| secondary spermatocytes (%) | 0.59 $\pm$ 0.41 | 0.69 $\pm$ 0.48 | 0.540 |
| round spermatids (%) | 35.79 $\pm$ 5.25 | 35.67 $\pm$ 3.94 | 0.696 |
| elongated spermatids (%) | 17.79 $\pm$ 3.86 | 17.38 $\pm$ 1.69 | 0.704 |
| spermatozoa (spermatic index, %) | 25.98 $\pm$ 6.44 | 25.93 $\pm$ 4.74 | 0.981 |
| *testicular indices* | | | |
| SEI (%) | *6.27 $\pm$ 3.18* | *4.24 $\pm$ 1.09* | *0.029* |
| SSEI | *5.16 $\pm$ 2.50* | *6.88 $\pm$ 1.82* | *0.010* |
| MI | 2.09 $\pm$ 0.47 | 2.04 $\pm$ 0.39 | 0.730 |
| ES/RS | 0.52 $\pm$ 0.14 | 0.50 $\pm$ 0.08 | 0.488 |
| ES/GC | 0.18 $\pm$ 0.04 | 0.17 $\pm$ 0.02 | 0.704 |
| RS/SC | *7.23 $\pm$ 3.42* | *9.51 $\pm$ 2.66* | *0.019* |
| ES/SC | *3.58 $\pm$ 1.96* | *4.62 $\pm$ 1.39* | *0.021* |
| GC/SC | *20.22 $\pm$ 9.63* | *26.51 $\pm$ 6.28* | *0.015* |

testicular indices tended to show a smaller inter-individual variability in the fallow deer from the high-energy diet group than those from the control group.

## 3.3. Blood plasma testosterone levels

There was no difference in the plasma testosterone levels between the fallow deer from the control and high-energy diet groups, although the latter showed the greatest values ($0.79 \pm 0.67$ ng ml$^{-1}$, $N = 15$ versus $1.08 \pm 0.70$ ng ml$^{-1}$, $N = 15$, for control and high-energy diet groups, respectively, $p = 0.116$).

## 3.4. Sperm concentration and acrosome integrity

All males had sperm cells in their epididymal caudae. There was only one fallow deer from the control group that showed a very low number of sperm cells; therefore, sperm concentration could not be assessed. The epididymal sperm concentration was 36.64% greater in the high-energy diet group than that of the control group ($p = 0.006$; figure 4a). There was no difference between treatments in the acrosome integrity ($96.94 \pm 1.79$% versus $96.47 \pm 1.60$%, for control and high-energy diet groups, respectively, $p = 0.440$).

## 3.5. Sperm morphometry and morphology

The proportion of each sperm structure in relation to the total sperm length in the whole fallow deer population ($N = 32$) was: head length, 12.23%; midpiece length, 20.04%; and principal piece plus terminal piece length, 67.74%. Interestingly, the fallow deer from the high-energy diet group showed longer sperm midpiece than those from the control group ($14.21 \pm 0.22$ versus $14.05 \pm 0.19$, respectively; $p = 0.029$). We did not find any difference between treatments in the rest of the sperm morphometry parameters evaluated ($p > 0.05$; table 4). On the other hand, the fallow deer from the

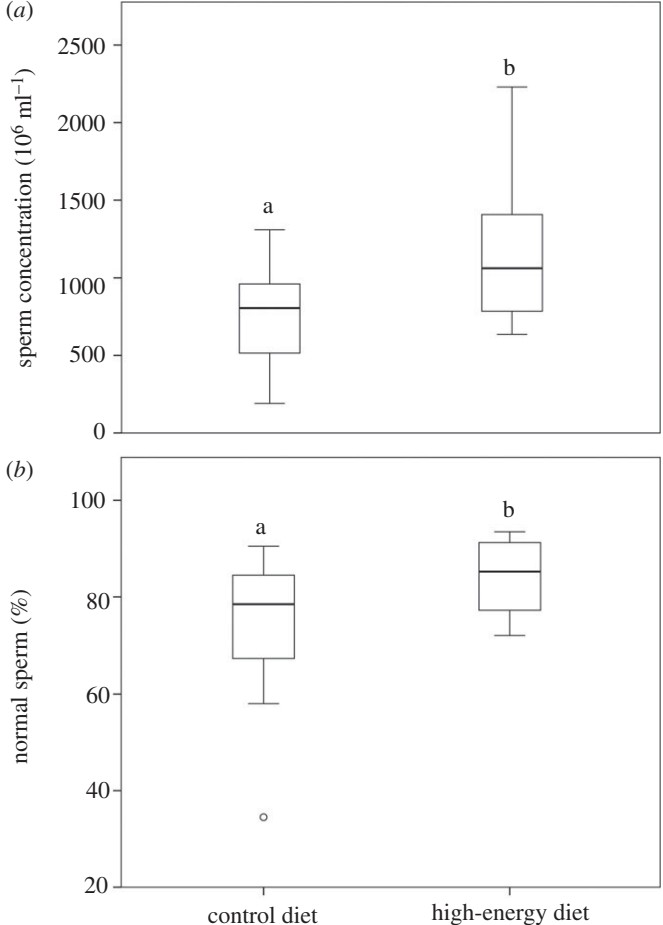

**Figure 4.** (*a*) Sperm concentration and (*b*) normal sperm morphology of fallow deer yearlings fed with pasture (control diet, for sperm concentration, $N = 15$; for sperm morphology, $N = 16$) or pasture with barley grain supplementation (high-energy diet, $N = 16$). Different letters above the box plots indicate significant differences between groups ($p < 0.05$).

**Table 4.** Sperm morphometry parameters of fallow deer yearlings fed with pasture (control diet, $N = 16$) or pasture with barley grain supplementation (high-energy diet, $N = 16$). Values are shown as mean $\pm$ s.d. Italics indicate significant differences between groups ($p < 0.05$).

| assessed parameters | control diet | high-energy diet | *p*-value |
|---|---|---|---|
| head width ($\mu$m) | 4.98 $\pm$ 0.12 | 5.05 $\pm$ 0.06 | 0.053 |
| head length ($\mu$m) | 8.61 $\pm$ 0.14 | 8.63 $\pm$ 0.16 | 0.811 |
| head area ($\mu$m$^2$) | 33.71 $\pm$ 1.10 | 34.22 $\pm$ 0.87 | 0.157 |
| head perimeter ($\mu$m) | 21.74 $\pm$ 0.33 | 21.85 $\pm$ 0.32 | 0.340 |
| midpiece length ($\mu$m) | *14.05 $\pm$ 0.19* | *14.21 $\pm$ 0.22* | *0.029* |
| principal plus terminal piece length ($\mu$m) | 47.84 $\pm$ 0.71 | 47.68 $\pm$ 0.62 | 0.482 |
| flagellum length ($\mu$m) | 61.89 $\pm$ 0.81 | 61.89 $\pm$ 0.73 | 0.991 |
| sperm length ($\mu$m) | 70.51 $\pm$ 0.86 | 70.52 $\pm$ 0.84 | 0.974 |

high-energy diet group had a higher percentage of sperm cells with a normal morphology compared to the control group ($p = 0.029$; figure 4*b*).

## 4. Discussion

In the present study, we found that barley grain supplemented diet administered prior to the breeding season promotes an increase in Sertoli cell functions and sperm midpiece size in fallow deer yearlings. To

the best of our knowledge, the effects of supplemented diet on the testicular cytology and the size of sperm components were not previously reported in any mammalian species. Moreover, we found that high-energy diet increases the body mass, testis mass, epididymal sperm concentration and percentage of normal sperm morphology. There was also a trend for the blood testosterone levels to be higher in the fallow deer fed with barley grain supplementation. The increase in sperm production and quality (i.e. normal sperm morphology) could be related to the greater functionality of Sertoli cells (spermatogenesis efficiency) found in the animals with a barley grain supplemented diet. Although our findings are clear and consistent, this study was performed on a single fallow deer population, which may limit the extrapolations of our results.

Overall, our findings confirm that high levels of nutritional intake increase the male reproductive performance, supporting what was previously found in other species (sheep [11,33]; rabbits [34]; bulls [35]). The underlying processes like spermatogenesis activity could be involved in the enhancement of reproductive function. The present study is, however, the first of its kind to evaluate the effects of nutrition on testicular functions by assessing the proportions of spermatogenic cell subtypes and several indices that quantify the spermatogenic activity and Sertoli cell functions using the FNAC technique. In this way, our study shows that high nutritional intake does not affect the proportion of any spermatogenic cell subtype, but it increases the Sertoli cell function and workload capacities, which may be responsible for the great sperm production and quality (normal sperm morphology) found in animals fed with barley grain. In rats, Melo *et al.* [36] found that protein restriction decreases the Sertoli cell efficiency (evaluated as the number of round spermatids per Sertoli cell), while it does not affect the mitotic and meiotic indices. In this way, barley grain represents an additional source of proteins and amino acids that enhance the spermatogenesis [37]. The chemical and estimated energetic values of diets also show that, despite the GE was overall similar between pasture and barley grain (about 18 MJ kg$^{-1}$ DM), the latter provides greater metabolizable energy, thanks to its high digestibility and DM content. The low amount of crude fibre, ADF and acid detergent lignin contained in the barley grain (more than four times less than those of pasture) contribute to the high digestibility of this feeding supplementation. In addition, the barley grain supplementation provides an extra source of macro- and micro-nutrients [19] that exert positive effects on spermatogenesis, sperm quality and fertility [37]. Thus, a large body of evidence shows that the restriction of nutrients intake negatively affects the male reproductive function both in invertebrates [38,39] and vertebrates [36,40]. Moreover, the great percentage of normal spermatozoa found in fallow deer fed with a high-energy diet might be a consequence of better functionality of Sertoli cells, which phagocytose apoptotic and degenerating germ cells [41,42]. Likewise, the SEI was lower in the fallow deer fed with barley grain, most likely because of their more efficient spermatogenic activity. Sertoli cells play a key role during the spermatogenesis and are important in determining testis size, spermatogenic activity, sperm output and quality [18,43–45]. Another pivotal function of the Sertoli cells concerns the nutritional support of the male germline, attained thanks to their ability to metabolize a large spectrum of substrates into relevant metabolites (e.g. lactate and pyruvate) for the developing germ cells [46]. In rats, Rato *et al.* [47] found that high-energy diet affects the testicular metabolism by increasing lactate content and lactate dehydrogenase activity. Given that the ruminant metabolism differs from that of non-ruminants in that, for instance, it largely depends on the microbial fermentation occurring in the forestomachs and little glucose is absorbed from the digestive tract [48], it remains to determine the effects of high-energy diet on the testicular metabolism in this mammalian suborder. In rams, Guan *et al.* [4] found that the dietary supplementation influences the Sertoli cell function (histological analyses), but it does not affect their number. By contrast, in underfed rams experiencing a reduction in testis mass and spermatogenesis, it seems that differentiation and maturation are reversed in the Sertoli cells, reducing their efficiency as supporters of the germ cells [49]. Because puberty entails the loss of Sertoli cell proliferative ability and the formation of the blood–testis barrier [50,51], it remains therefore to be tested whether in fallow deer yearlings, the high nutritional intake may have influenced the Sertoli cell number or its maturational development during puberty.

Another interesting result of this study is that the dietary supplementation did not affect the percentage of the spermatogenic cell subtypes and nor the indices of germ cell loss. The lack of effect on these parameters can be due to the fact that control diet was able to supply most of the energetic needs required for reproductive functions, given that the mean body mass and testes mass of fallow deer in our study are similar to the values previously reported [12]. On this basis, none of the animals in our study seemed to be therefore overweight or undernourished. By contrast, it is known that both obesity and undernourishment negatively affect the male reproduction by impairing the

spermatogenesis [33,52]. Our results also show that the percentage of spermatozoa (i.e. spermatic index) did not vary between treatments, whereas the epididymal sperm concentration was increased in the group with barley grain supplementation. Similarly, in beef [8] and dairy [10,53] bulls, high-energy diets did not affect the daily sperm production in spite of bigger testis size. Moreover, in agreement with our results, Dance et al. [10] found that also the epididymal sperm reserves were increased by the high-energy diet. Our findings can be explained by the fact that the spermatic index and the epididymal sperm concentration quantify different aspects of sperm production. While the spermatic index indicates the percentage of spermatozoa over the whole germ cell population, the epididymal sperm concentration estimates the epididymides' storage capacity, which depends on factors like the phase of the breeding season and ejaculatory frequency, among others. One plausible explanation is that the dietary supplementation may have fastened the onset of puberty, leading to greater epididymal sperm reservoirs. To test this hypothesis, further studies should require repeated sample collections from the same individual throughout the breeding season. Another explanation is that the dietary supplementation did not influence the proportion of spermatozoa per se, but rather increased the overall germ cell population, as confirmed by the great number of germ cells per Sertoli cell and epididymal sperm concentration. We also found that high-energy diet tended to increase the blood testosterone levels, which may have contributed to a possible earlier onset of puberty. An increase in the testosterone levels due to a modified diet has been previously reported in other domestic ungulates like rams [54] and bulls [10]. However, the role of other hormones like the follicle-stimulating hormone, thyroid hormone and insulin should be considered in future studies, given their relevance in spermatogenesis and Sertoli cell functions [55].

One striking result from this study is that high-energy diet increases the size of the sperm midpiece. The sperm mitochondria are helically wrapped around the flagellum (mitochondrial sheath) forming the midpiece and are considered the cell powerhouse and biomarkers of sperm quality and fertilization ability [56,57]. Because the mitochondria contribute to the energy supply (ATP) required for the sperm motility, an increase in the sperm midpiece may indeed enhance sperm velocity and lifespan. Nevertheless, sperm cells show different ways to obtain ATP [58]. While sperm cells from some species mostly use the ATP generated in the mitochondria by oxidative phosphorylation (OXPHOS), others obtain most of this energy from the principal piece via glycolysis [57]. The differences among species in the way to obtain the main source of ATP (OXPHOS versus glycolysis) used for sperm motility are not yet elucidated. The relative length of the sperm midpiece with respect to the sperm size could be associated with the way in which sperm mostly obtain the ATP required for motility (relatively long midpiece may suggest OXPHOS as a main source of ATP, while relatively short midpiece may suggest glycolysis as a main source of ATP). To date, only a few studies have explored the relationship between nutrition and sperm morphometry. In lizards (Anolis sagrei) and red squirrels (Tamiasciurus hudsonicus), in which the sperm midpiece represents less than 5% of the total sperm length, a food restriction and a poor body condition are associated with an increase in the sperm midpiece ([59,60]; but see [61]). Our results show that sperm midpiece in fallow deer constitutes 20% of the total sperm length and may indicate that in this species the most important source of ATP used for sperm motility comes via OXPHOS. Taken together, these findings suggest that the diet or body condition may influence changes in the size of sperm structures depending on the source of energy employed for sperm movement across animal species. On the other hand, a high dietary protein intake in rats promotes an increase in the levels of the enzyme CPS1 (involved in protein and nitrogen metabolism) that, in turn, induces an increase in the size and number of mitochondria in hepatocytes [62,63]. Therefore, a higher protein intake (provided by the barley grain supplementation) in fallow deer may be associated with increased levels or activity of some mitochondrial enzymes related to the sperm metabolism [64] that could promote an enlargement of the sperm midpiece. Another plausible explanation of this phenomenon may concern the tendency of higher testosterone levels found in the animals with high-energy diet. In this way, Immler et al. [65] found that higher testosterone levels are associated with longer sperm midpiece in birds. The mechanisms by which nutrition influences the size of sperm midpiece are still unclear and deserve further investigation.

## 5. Conclusion

Our results show that dietary supplementation with whole barley grain enhances the male reproductive traits (e.g. greater testes mass, sperm production and percentage of normal sperm morphology) in fallow deer yearlings. The enhancement of male reproductive performance might

have been a consequence of an earlier onset of puberty and is probably related to the improved spermatogenic functions (i.e. Sertoli cell function and workload capacity). Moreover, high-energy diet increases the sperm midpiece, but the underlying mechanisms of this phenomenon and their implications on sperm function are unknown at present. The results of this study provide new insights into the importance of nutrition in the male sexual development and might be useful for better practices of livestock management in seasonal breeders.

Ethics. All experimental procedures were approved by the Animal Care Committee of the Institute of Animal Science (IAS, IACUC No. 60444/2011-MZE-17214).

Data accessibility. All data generated or analysed during this study are included in this published article and its electronic supplementary material.

Authors' contributions. J.L.R.-S. conceived and designed the experiments, estimated the energetic value of the diet, performed the laboratory analysis, analysed the data and wrote the manuscript; R.K. conceived and designed the experiments, and performed the fieldwork; E.P. conceived and designed the experiments, estimated the energetic value of the diet, performed the laboratory analysis, analysed the data and wrote the manuscript. All authors gave final approval for publication.

Competing interests. The authors declare no competing interest.

Funding. The study was supported by the Ministry of Agriculture of the Czech Republic (grant no. MZE-RO0718) and by the Faculty of Tropical AgriSciences (grant no. IGA-20175014).

Acknowledgements. The authors are grateful to the farmer Pavel Friedberger and his family to allow experimentation and providing samples and to Filip Jančík for chemical analyses of feed.

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
