## [Reviewer comments · Royal Society Open Science]

Review History

RSOS-181972.R0 (Original submission)

Review form: Reviewer 1

Is the manuscript scientifically sound in its present form?

Yes

Are the interpretations and conclusions justified by the results?

Yes

Is the language acceptable?

Yes

Is it clear how to access all supporting data?

Not Applicable

Do you have any ethical concerns with this paper?

No

Have you any concerns about statistical analyses in this paper?

No

Recommendation?

Major revision is needed (please make suggestions in comments)

Comments to the Author(s)

Comments to the Authors:

The authors explore the role of diet in shaping the sexual development in male fallow deer. Supplemental feed and pasture composition were determined using several nutritional analyses. Moreover, metrics for growth, testis mass, spermatogenic activity, sperm traits, and testosterone levels were determined. The authors found that males grew more, had higher testicular mass, and also had higher sperm quality when provided supplemental grains.

This study investigates an interesting and relatively understudied question. Many novel aspects of reproductive responses to diet have been explored in this study. My major concerns pertain to the use of energy as the primary indicator of nutrient quality without actually providing estimates for dietary energetic differences and the potentially strong alternative explanations driving the differences in response variables. The nutritive composition of the diet is really only cited in a single sentence (Lines 92-93), while later in the Discussion section only the protein and energetic content seems to be robustly considered. The protein content was slightly higher in the grass and the fat content was slightly higher in the barley grain. However, the dry matter content was over twice as great in the barley, while the fiber and lignin content were approximately three times as high in the grass. These factors might contribute to digestibility and assimilation differences/costs between the diets, which could be interacting with the response variables (in addition to energy intake). The study also seems to have been conducted on single population (divided into a control and treatment) during a single season, with most sample collection occurring at the terminus, which perhaps limits extrapolation of the findings.

Specific Comments:

1. Line 71-73: This statement seems to offer one of the strongest nutritive explanations for why supplementation with barley might increase fertility in male fallow deer. However, essential amino acids did not seem to be quantified in this study. Other nutritive explanations for the advantages of supplementation (digestibility, higher fat content, etc.) might give broader perspective to the study from the dietary data provided.
2. Lines 98-104: Was nutrient composition of barley grain determined using the same analyses as pasture or were other methods used to infer?
3. Lines 106-108: This seems like sufficient justification for using an alternative to randomized sample collection.
4. Lines 175-177: It looks like the deer were initially size-matched for the treatments, which I think was well recognized to be a necessary part of the experimental design.
5. Lines 189-193 and 215-217: These seemed to be the most significant experimental findings. Many of the other response variables did not seem to differ (e.g. testosterone, acrosomal status, and several aspects of sperm morphology).
6. Lines 230- 253: The authors seem well versed in the function and production of male gametes, however, the link to diet is not yet well-elucidated. The only direct nutritional explanation is from the early reference to essential amino acids present (but not actually measured in this study) in the grains.
7. Lines 291-293: These lines provide a citation to the broader diversity of animals in the literature, which is relatively sparse up to this point in the paper. However, the animals in the

current study seemed to be neither food restricted nor in poor body condition given the gain in mass and at least some equivalent aspects of fertility in both treatment and control.

8. Lines 300- 302: It is difficult to assess this claim because Table 1 appears to show higher protein available for consumer intake in the grass compared to grain.

9. Line 498: At present, Table 1 only seems to be provided as background rather than as a supporting piece of evidence. This table is only cited once (within the Methods) and only protein and perhaps fat are alluded to. It is not yet clear why other nutrient analyses in the table were conducted and if these biochemicals might also contribute to the observed differences in the response variables. The arguments for higher-energy diet contributing to increased spermatogenic function and sperm midpiece length would also be strengthened if calorimetry findings or standard energy conversion estimates were to be included for the diets (e.g. protein=17 kJ/g, and fat=37 kJ/g; Raubenheimer and Rothman. 2013. Nutritional ecology of entomophagy in humans and other primates. *Annu Rev Entomol*).

Review form: Reviewer 2

Is the manuscript scientifically sound in its present form?

No

Are the interpretations and conclusions justified by the results?

Yes

Is the language acceptable?

Yes

Is it clear how to access all supporting data?

No

Do you have any ethical concerns with this paper?

No

Have you any concerns about statistical analyses in this paper?

No

Recommendation?

Major revision is needed (please make suggestions in comments)

Comments to the Author(s)

In this work, the authors elicit the influence of an energy-rich diet on the onset of sexual maturation of fallow deer (*Dama dama*). To achieve this, the authors randomly allocated 32 pre-pubertal fallow deer to two groups: one fed with pasture and one supplemented with barley. Six months later, the animals were sacrificed and biometric parameters, serum testosterone, sperm parameters, spermatogenic function and acrosomal status were assessed. Overall, the authors report improvements in sperm parameters and reproductive function, compatible to an earlier onset of sexual maturation in the group fed with pasture supplemented with barley.

This paper presents some results that explore the potential commercial viability of fallow deer, based on an adequate methodology. The evaluated parameters are significant according to the aims of the work, and variables are properly defined. Besides, considering the amount of collected data and its further treatment, the reported results seem reliable.

Nevertheless, there are some major concerns regarding the way the authors present their data,

and some gaps that must be filled in order to draw more consistent conclusions.

1. The authors have not identified the most critical aim of their work. The most evident conclusion of this work is that a high-energy diet anticipates the puberty onset on fallow deer. The authors do not present any conclusive proof that it was diet, and not sexual development during puberty, the responsible for the improvement in sperm parameters. In fact, even the bibliography cited by the authors raise the idea that it was the maturation state, likely due to nutritional state, the main differentiating factor between the two groups of deer. Therefore, the authors must rewrite the paper around this aim, which will forcibly mean changes in all sections, including title.
2. The nomenclature of scientific terms is incoherent throughout the paper, rendering it unclear. As example, the nomenclature used for germ cell development phases is not coherent between Figure 1, Table 2 and the manuscript (the authors use round spermatocytes in text, then in table 2 they present counts for primary and secondary spermatocytes). Additionally, the authors should clearly define what are "sperm parameters", "spermatogenic function" and "acrosomal status".
3. Figure 1 is an essential figure in the context of the work but needs further refinement. It should show a wider field, or the authors could supplement it with an image of lower magnification, in order to show the seminiferous tubule structure and integrity.
4. During discussion, the authors overestimate their observations based upon inadequate bibliography. It is not comparable sperm mid-piece size of lizards with deer, particularly if no measurement of mitochondria number and fitness was performed. Besides, it is not acceptable to relate that eventual gain to a diet effect, based upon data draw from a study in rat liver. If the authors want to make this connection, they must present data on mitochondrial number and function in spermatozoa's midpiece.
5. Since the authors collected serum of the animals, they should quantify other hormones of the reproductive axis, in order to further characterize the maturation phase of the deer at sacrifice.
6. Although the study focuses on fallow deer, the authors must not disregard works published in other mammal species, notably rodents and humans, where it is stated that high-energy diets lead to impaired sperm parameters and poorer fertility outcomes.
7. I would not draw conclusions out of non-significant results as if there is a difference (e.g. testosterone levels).

Additionally, there are minor issues that should be addressed when rewriting the article.

1. Please, review the superscripts in the boxplots and state what "a" and "b" stand for.
2. I advise the authors to look at the following bibliography to improve their work, especially regarding germ cell - Sertoli cell cooperation: PMID 23495257, 29453570; ISBN 978-3-319-19791-3.

Decision letter (RSOS-181972.R0)

15-Mar-2019

Dear Dr Pintus,

The editors assigned to your paper ("High-energy diet enhances spermatogenic function and increases sperm midpiece length in fallow deer (*Dama dama*)") have now received comments from reviewers. We would like you to revise your paper in accordance with the referee and Associate Editor suggestions which can be found below (not including confidential reports to the Editor). Please note this decision does not guarantee eventual acceptance.

Please submit a copy of your revised paper before 07-Apr-2019. Please note that the revision deadline will expire at 00.00am on this date. If we do not hear from you within this time then it

will be assumed that the paper has been withdrawn. In exceptional circumstances, extensions may be possible if agreed with the Editorial Office in advance. We do not allow multiple rounds of revision so we urge you to make every effort to fully address all of the comments at this stage. If deemed necessary by the Editors, your manuscript will be sent back to one or more of the original reviewers for assessment. If the original reviewers are not available, we may invite new reviewers.

- Data accessibility

If you wish to submit your supporting data or code to Dryad (<http://datadryad.org/>), or modify your current submission to dryad, please use the following link:
<http://datadryad.org/submit?journalID=RSOS&manu=RSOS-181972>

- Competing interests

- Authors' contributions

- Acknowledgements

- Funding statement

on behalf of Dr Punidan Jeyasingh (Associate Editor) and Professor Kevin Padian (Subject Editor)
openscience@royalsociety.org

Associate Editor's comments (Dr Punidan Jeyasingh):

This manuscript reporting effects of an experimental high-energy diet on sperm morphology and functionality in a livestock ruminant was evaluated by two experts. I apologize for the long delay because I had a hard time finding suitable reviewers. The experts liked the work, and clearly see its importance. Nevertheless, they have raised some serious issues. Of particular note is the lack of data on the nutritional parameters of the diet. This information must be more clearly furnished, and the results discussed in this light. Clearly, dietary changes are bound to have myriad physiological implications, making the links between diet and sperm traits, as claimed in this study, rather obscure. I believe the manuscript can be improved along these lines. Moreover, the authors appear to have ignored similar work in other taxa which can be leveraged to strengthen inferences. I felt the reviews were fair and constructive. With much gratitude to the expert reviewers, I wish the authors the best in addressing these comments and improving their manuscript.

Comments to Author:

Reviewers' Comments to Author:

Reviewer: 1

Comments to the Author(s)

Comments to the Authors:

The authors explore the role of diet in shaping the sexual development in male fallow deer. Supplemental feed and pasture composition were determined using several nutritional analyses. Moreover, metrics for growth, testis mass, spermatogenic activity, sperm traits, and testosterone

levels were determined. The authors found that males grew more, had higher testicular mass, and also had higher sperm quality when provided supplemental grains.

This study investigates an interesting and relatively understudied question. Many novel aspects of reproductive responses to diet have been explored in this study. My major concerns pertain to the use of energy as the primary indicator of nutrient quality without actually providing estimates for dietary energetic differences and the potentially strong alternative explanations driving the differences in response variables. The nutritive composition of the diet is really only cited in a single sentence (Lines 92-93), while later in the Discussion section only the protein and energetic content seems to be robustly considered. The protein content was slightly higher in the grass and the fat content was slightly higher in the barley grain. However, the dry matter content was over twice as great in the barley, while the fiber and lignin content were approximately three times as high in the grass. These factors might contribute to digestibility and assimilation differences/costs between the diets, which could be interacting with the response variables (in addition to energy intake). The study also seems to have been conducted on single population (divided into a control and treatment) during a single season, with most sample collection occurring at the terminus, which perhaps limits extrapolation of the findings.

Specific Comments:

1. Line 71-73: This statement seems to offer one of the strongest nutritive explanations for why supplementation with barley might increase fertility in male fallow deer. However, essential amino acids did not seem to be quantified in this study. Other nutritive explanations for the advantages of supplementation (digestibility, higher fat content, etc.) might give broader perspective to the study from the dietary data provided.
2. Lines 98-104: Was nutrient composition of barley grain determined using the same analyses as pasture or were other methods used to infer?
3. Lines 106-108: This seems like sufficient justification for using an alternative to randomized sample collection.
4. Lines 175-177: It looks like the deer were initially size-matched for the treatments, which I think was well recognized to be a necessary part of the experimental design.
5. Lines 189-193 and 215-217: These seemed to be the most significant experimental findings. Many of the other response variables did not seem to differ (e.g. testosterone, acrosomal status, and several aspects of sperm morphology).
6. Lines 230- 253: The authors seem well versed in the function and production of male gametes, however, the link to diet is not yet well-elucidated. The only direct nutritional explanation is from the early reference to essential amino acids present (but not actually measured in this study) in the grains.
7. Lines 291-293: These lines provide a citation to the broader diversity of animals in the literature, which is relatively sparse up to this point in the paper. However, the animals in the current study seemed to be neither food restricted nor in poor body condition given the gain in mass and at least some equivalent aspects of fertility in both treatment and control.
8. Lines 300- 302: It is difficult to assess this claim because Table 1 appears to show higher protein available for consumer intake in the grass compared to grain.
9. Line 498: At present, Table 1 only seems to be provided as background rather than as a supporting piece of evidence. This table is only cited once (within the Methods) and only protein and perhaps fat are alluded to. It is not yet clear why other nutrient analyses in the table were conducted and if these biochemicals might also contribute to the observed differences in the response variables. The arguments for higher-energy diet contributing to increased spermatogenic function and sperm midpiece length would also be strengthened if calorimetry findings or standard energy conversion estimates were to be included for the diets (e.g. protein=17 kJ/g, and fat=37 kJ/g; Raubenheimer and Rothman. 2013. Nutritional ecology of entomophagy in humans and other primates. *Annu Rev Entomol*).

Reviewer: 2

Comments to the Author(s)

In this work, the authors elicit the influence of an energy-rich diet on the onset of sexual maturation of fallow deer (*Dama dama*). To achieve this, the authors randomly allocated 32 pre-pubertal fallow deer to two groups: one fed with pasture and one supplemented with barley. Six months later, the animals were sacrificed and biometric parameters, serum testosterone, sperm parameters, spermatogenic function and acrosomal status were assessed. Overall, the authors report improvements in sperm parameters and reproductive function, compatible to an earlier onset of sexual maturation in the group fed with pasture supplemented with barley.

This paper presents some results that explore the potential commercial viability of fallow deer, based on an adequate methodology. The evaluated parameters are significant according to the aims of the work, and variables are properly defined. Besides, considering the amount of collected data and its further treatment, the reported results seem reliable.

Nevertheless, there are some major concerns regarding the way the authors present their data, and some gaps that must be filled in order to draw more consistent conclusions.

1. The authors have not identified the most critical aim of their work. The most evident conclusion of this work is that a high-energy diet anticipates the puberty onset on fallow deer. The authors do not present any conclusive proof that it was diet, and not sexual development during puberty, the responsible for the improvement in sperm parameters. In fact, even the bibliography cited by the authors raise the idea that it was the maturation state, likely due to nutritional state, the main differentiating factor between the two groups of deer. Therefore, the authors must rewrite the paper around this aim, which will forcibly mean changes in all sections, including title.

2. The nomenclature of scientific terms is incoherent throughout the paper, rendering it unclear. As example, the nomenclature used for germ cell development phases is not coherent between Figure 1, Table 2 and the manuscript (the authors use round spermatocytes in text, then in table 2 they present counts for primary and secondary spermatocytes). Additionally, the authors should clearly define what are "sperm parameters", "spermatogenic function" and "acrosomal status".

3. Figure 1 is an essential figure in the context of the work but needs further refinement. It should show a wider field, or the authors could supplement it with an image of lower magnification, in order to show the seminiferous tubule structure and integrity.

4. During discussion, the authors overestimate their observations based upon inadequate bibliography. It is not comparable sperm mid-piece size of lizards with deer, particularly if no measurement of mitochondria number and fitness was performed. Besides, it is not acceptable to relate that eventual gain to a diet effect, based upon data drawn from a study in rat liver. If the authors want to make this connection, they must present data on mitochondrial number and function in spermatozoa's midpiece.

5. Since the authors collected serum of the animals, they should quantify other hormones of the reproductive axis, in order to further characterize the maturation phase of the deer at sacrifice.

6. Although the study focuses on fallow deer, the authors must not disregard works published in other mammal species, notably rodents and humans, where it is stated that high-energy diets lead to impaired sperm parameters and poorer fertility outcomes.

7. I would not draw conclusions out of non-significant results as if there is a difference (e.g. testosterone levels).

Additionally, there are minor issues that should be addressed when rewriting the article.

1. Please, review the superscripts in the boxplots and state what "a" and "b" stand for.

2. I advise the authors to look at the following bibliography to improve their work, especially regarding germ cell – Sertoli cell cooperation: PMID 23495257, 29453570; ISBN 978-3-319-19791-3.

Author's Response to Decision Letter for (RSOS-181972.R0)

See Appendix A.

RSOS-181972.R1 (Revision)

Review form: Reviewer 1

Is the manuscript scientifically sound in its present form?

Yes

Are the interpretations and conclusions justified by the results?

Yes

Is the language acceptable?

Yes

Is it clear how to access all supporting data?

Yes

Do you have any ethical concerns with this paper?

No

Have you any concerns about statistical analyses in this paper?

No

Recommendation?

Accept as is

Comments to the Author(s)

Overall, I find that the authors have comprehensively addressed my concerns from the previous draft. The authors have provided additional clarification and seemed to effectively revise paragraph structure within the Methods. Inclusion of Table 2 (Lines 589-592) sufficiently reports the energetics of pasture versus the supplemental diet. This is exemplified through partitioning of the gross and digestible/metabolisable components, which seems to support a more robust series of explanations in the revised Discussion section. Further, the authors have integrated consideration of scope to their findings (Lines 242-243, 305-307) while also providing a more thorough consideration of the literature base (throughout Discussion) necessary to compel future study in this understudied research area.

Review form: Reviewer 2

Is the manuscript scientifically sound in its present form?

Yes

Are the interpretations and conclusions justified by the results?

Yes

Is the language acceptable?

Yes

Is it clear how to access all supporting data?

Not Applicable

Do you have any ethical concerns with this paper?

No

Have you any concerns about statistical analyses in this paper?

No

Recommendation?

Accept with minor revision (please list in comments)

Comments to the Author(s)

In this work, the authors elicit the influence of an energy-rich diet on the onset of sexual maturation of fallow deer (*Dama dama*). To achieve this, the authors randomly allocated 32 pre-pubertal fallow deer to two groups: one fed with pasture and one supplemented with barley. Six months later, the animals were sacrificed and biometric parameters, serum testosterone, sperm parameters, spermatogenic function and acrosomal status were accessed. Overall, the authors report improvements in sperm parameters and reproductive function, compatible to an earlier onset of sexual maturation in the group fed with pasture supplemented with barley.

After reading their revised paper, and considering all the changes in result of the reviewer's criticism, the outcome is a clearly an improved manuscript. I would like to particularly address the improvements in discussion, which now reflect a sounder rational and easier reading.

I consider that my central criticism regarding this paper was addressed. As I pointed out before, one of the limitations of this work is the impossibility to separate developmental and dietary factors as the driving force of the male fertility parameters. This concern is addressed by the authors (lines 317-318), but a plethora of other methodologies could suit this problem in further works. Therefore, my advice is to restrict your conclusions to "young adult fallow deer", or a more suitable veterinarian term to describe the developmental phase of animals included in this study. This change should also be reflected in title.

Lastly, I call the author's attention to a minor aspect. In pages 71 and 72, the significance marks are not clear. The authors should state what the lower-case letters "a" and "b" stand for in the image subtitle. I understand they denote significance, but does "a" and "b" stand for different levels of significance? Or does it stand for group-group comparisons?

Decision letter (RSOS-181972.R1)

07-May-2019

Dear Dr Pintus:

On behalf of the Editors, I am pleased to inform you that your Manuscript RSOS-181972.R1 entitled "High-energy diet enhances spermatogenic function and increases sperm midpiece length in fallow deer (*Dama dama*)" has been accepted for publication in Royal Society Open

Science subject to minor revision in accordance with the referee suggestions. Please find the referees' comments at the end of this email.

The reviewers and Subject Editor have recommended publication, but also suggest some minor revisions to your manuscript. Therefore, I invite you to respond to the comments and revise your manuscript.

- Ethics statement

- Data accessibility

If you wish to submit your supporting data or code to Dryad (<http://datadryad.org/>), or modify your current submission to dryad, please use the following link:
<http://datadryad.org/submit?journalID=RSOS&manu=RSOS-181972.R1>

- Competing interests

- Authors' contributions

- Acknowledgements

- Funding statement

Because the schedule for publication is very tight, it is a condition of publication that you submit the revised version of your manuscript before 16-May-2019. Please note that the revision deadline will expire at 00.00am on this date. If you do not think you will be able to meet this date please let me know immediately.

on behalf of Dr Punidan Jeyasingh (Associate Editor) and Kevin Padian (Subject Editor)
openscience@royalsociety.org

Associate Editor Comments to Author (Dr Punidan Jeyasingh):

I thank the authors for a thoroughly revised version of the manuscript. This revision was assessed by the original reviewers, and both are clearly happy with it. With much gratitude to the reviewers, I am happy to recommend this manuscript for publication. One of the reviewers picked up a couple of important issues that should be addressed before the manuscript can be accepted for publication.

Reviewer comments to Author:

Reviewer: 2

Comments to the Author(s)

In this work, the authors elicit the influence of an energy-rich diet on the onset of sexual maturation of fallow deer (*Dama dama*). To achieve this, the authors randomly allocated 32 pre-pubertal fallow deer to two groups: one fed with pasture and one supplemented with barley. Six months later, the animals were sacrificed and biometric parameters, serum testosterone, sperm parameters, spermatogenic function and acrosomal status were assessed. Overall, the authors report improvements in sperm parameters and reproductive function, compatible with an earlier onset of sexual maturation in the group fed with pasture supplemented with barley.

After reading their revised paper, and considering all the changes in result of the reviewer's criticism, the outcome is a clearly improved manuscript. I would like to particularly address the improvements in discussion, which now reflect a sounder rationale and easier reading. I consider that my central criticism regarding this paper was addressed. As I pointed out before, one of the limitations of this work is the impossibility to separate developmental and dietary factors as the driving force of the male fertility parameters. This concern is addressed by the authors (lines 317-318), but a plethora of other methodologies could suit this problem in further works. Therefore, my advice is to restrict your conclusions to "young adult fallow deer", or a more suitable veterinarian term to describe the developmental phase of animals included in this study. This change should also be reflected in the title.

Lastly, I call the author's attention to a minor aspect. In pages 71 and 72, the significance marks are not clear. The authors should state what the lower-case letters "a" and "b" stand for in the image subtitle. I understand they denote significance, but does "a" and "b" stand for different levels of significance? Or does it stand for group-group comparisons?

Reviewer: 1

Comments to the Author(s)

Overall, I find that the authors have comprehensively addressed my concerns from the previous draft. The authors have provided additional clarification and seemed to effectively revise paragraph structure within the Methods. Inclusion of Table 2 (Lines 589-592) sufficiently reports the energetics of pasture versus the supplemental diet. This is exemplified through partitioning of the gross and digestible/metabolisable components, which seems to support a more robust series of explanations in the revised Discussion section. Further, the authors have integrated consideration of scope to their findings (Lines 242-243, 305-307) while also providing a more thorough consideration of the literature base (throughout Discussion) necessary to compel future study in this understudied research area.

Author's Response to Decision Letter for (RSOS-181972.R1)

See Appendix B.

Decision letter (RSOS-181972.R2)

13-May-2019

Dear Dr Pintus,

I am pleased to inform you that your manuscript entitled "High-energy diet enhances spermatogenic function and increases sperm midpiece length in fallow deer (*Dama dama*) yearlings" is now accepted for publication in Royal Society Open Science.

on behalf of Dr Punidan Jeyasingh (Associate Editor) and Kevin Padian (Subject Editor)
openscience@royalsociety.org

Appendix A

Associate Editor

We thank Associate Editor for the time taken to review our manuscript. We really appreciate his comments and suggestion, which helped us to improve our manuscript. Line numbers, where changes were made, refer to the clean version of the manuscript.

Associate Editor's comments (Dr Punidan Jeyasingh):

This manuscript reporting effects of an experimental high-energy diet on sperm morphology and functionality in a livestock ruminant was evaluated by two experts. I apologize for the long delay because I had a hard time finding suitable reviewers. The experts liked the work, and clearly see its importance. Nevertheless, they have raised some serious issues. Of particular note is the lack of data on the nutritional parameters of the diet. This information must be more clearly furnished, and the results discussed in this light. Clearly, dietary changes are bound to have myriad physiological implications, making the links between diet and sperm traits, as claimed in this study, rather obscure. I believe the manuscript can be improved along these lines. Moreover, the authors appear to have ignored similar work in other taxa which can be leveraged to strengthen inferences. I felt the reviews were fair and constructive. With much gratitude to the expert reviewers, I wish the authors the best in addressing these comments and improving their manuscript.

As suggested by Reviewer 1, we have included in the manuscript a new table (Table 2) providing several nutritional parameters of the diets (i.e. organic matter digestibility, energy digestibility, gross energy, digestible energy, and metabolisable energy). On the basis of these findings, we discussed our results from a deeper and broader perspective (lines 255-265). Moreover, we have included several new references, including those recommended by Reviewer 2. As suggested by both Reviewers, as limitations of our study, we have stated in the manuscript that i) the study was performed on a single population of fallow deer (as indicated by Reviewer 1) (lines 242-243) and ii) that additional hormonal analysis should be considered in future studies (as indicated by Reviewer 2) (lines 313-315).

We hope that the revised version of the manuscript is now clearly presented and technically sound.

Reviewer: 1

We thank Reviewer 1 for the time taken to review our manuscript. We really appreciate his/her comments and suggestion, which helped us to improve our manuscript. Line numbers, where changes were made, refer to the clean version of the manuscript.

Comments to the Authors:

The authors explore the role of diet in shaping the sexual development in male fallow deer. Supplemental feed and pasture composition were determined using several nutritional analyses. Moreover, metrics for growth, testis mass, spermatogenic activity, sperm traits, and testosterone levels were determined. The authors found that males grew more, had higher testicular mass, and also had higher sperm quality when provided supplemental grains.

This study investigates an interesting and relatively understudied question. Many novel aspects of reproductive responses to diet have been explored in this study. My major concerns pertain to the use of energy as the primary indicator of nutrient quality without actually providing estimates for dietary energetic differences and the potentially strong alternative explanations driving the differences in response variables. The nutritive composition of the diet is really only cited in a single sentence (Lines 92-93), while later in the Discussion section only the protein and energetic content seems to be robustly considered. The protein content was slightly higher in the grass and the fat content was slightly higher in the barley grain. However, the dry matter content was over twice as great in the barley, while the fiber and lignin content were approximately three times as high in the grass. These factors might contribute to digestibility and assimilation differences/costs between the diets, which could be interacting with the response variables (in addition to energy intake). The study also seems to have been conducted on single population (divided into a control and treatment) during a single season, with most sample collection occurring at the terminus, which perhaps limits extrapolation of the findings.

As suggested by Reviewer 1, we have included in the manuscript a new table (Table 2) providing several nutritional parameters of the diets (i.e. organic matter digestibility, energy digestibility, gross energy, digestible energy, and metabolisable energy). On the basis of these findings, we include a new paragraph in the discussion section (lines 255-265). We would like also to point out that in our study the samples (testes) were collected during the breeding season (autumn-October) because during the rest of the year fallow deer show little or no sperm production. Nevertheless, because our study was conducted on a single population, we stated that this factor may limit the extrapolation of our findings (lines 242-243).

Specific Comments:

1. Line 71-73: This statement seems to offer one of the strongest nutritive explanations for why supplementation with barley might increase fertility in male fallow deer. However, essential amino acids did not seem to be quantified in this study. Other nutritive explanations for the advantages of supplementation (digestibility, higher fat content, etc.) might give broader perspective to the study from the dietary data provided.

As Reviewer 1 suggested, the estimation of energetic value of the diets (Table 2) helped us to discuss our findings in a broader perspective that consider the composition and energetic value of the feeds (lines 255-265).

2. Lines 98-104: Was nutrient composition of barley grain determined using the same analyses as pasture or were other methods used to infer?

Thank you for your comment. Yes, the nutrient composition of barley grain was determined using the same analyses as the pasture. We have included this information in the manuscript (lines 102).

3. Lines 106-108: This seems like sufficient justification for using an alternative to randomized sample collection.

Samples were processed using the same order of slaughtering in order to minimize the interval between death and sperm analyses. In this way, all samples were processed approximately after the same time interval after animal's death. Moreover, the order of slaughtering (pasture vs barley grain) was randomly established and the researchers who perform the testicular and sperm analyses were blind about the group assignment of the animals. This information has been included in the manuscript (lines 121-123).

4. Lines 175-177: It looks like the deer were initially size-matched for the treatments, which I think was well recognized to be a necessary part of the experimental design.

At the beginning of the experiment, the deer were allocated in two groups and we checked that there were no differences in the body mass between treatments.

5. Lines 189-193 and 215-217: These seemed to be the most significant experimental findings. Many of the other response variables did not seem to differ (e.g. testosterone, acrosomal status, and several aspects of sperm morphology).

We agree with you. The effects of high energy diet on the quantitative assessment of spermatogenesis by testicular cytology together with the increase of the sperm midpiece length are the most relevant findings of this study.

6. Lines 230- 253: The authors seem well versed in the function and production of male gametes, however, the link to diet is not yet well-elucidated. The only direct nutritional explanation is from the early reference to essential amino acids present (but not actually measured in this study) in the grains.

As suggested by Reviewer 1, the estimation of the energetic values of pasture and barley grain helped us to discuss our finding in a broader perspective (lines 255-265). Despite the gross energy of barley and pasture were similar, the higher digestibility of the barley grain (i.e. low fiber amount and high dry matter content) contributes to its higher digestible and metabolisable energy compared to those of pasture.

7. Lines 291-293: These lines provide a citation to the broader diversity of animals in the literature, which is relatively sparse up to this point in the paper. However, the animals in the current study seemed to be neither food restricted nor in poor body condition given the gain in mass and at least some equivalent aspects of fertility in both treatment and control.

Thank you for your comments and observations. We are aware that some of the citations within these lines are more related with food restriction and poor body condition. The current literature about the effects of nutrition in the size of sperm structures is extremely scarce. For this reason, we think that the cited references here are useful to provide a plausible explanation to our findings. However, we have included in the manuscript that, in our study, none of the fallow deer seemed to be undernourished, given that mean body mass was similar to the values previously reported by

Chapman and Chapman (1970) (lines 291-292). In addition, we included new references about the effect of nutrition on male reproduction in other species (lines 265 and 294).

8. Lines 300- 302: It is difficult to assess this claim because Table 1 appears to show higher protein available for consumer intake in the grass compared to grain.

Yes, pasture shows slightly higher protein content compared to the barley grain. Nevertheless, we have to take into account that all animals in this study were fed on pasture. The barley grain represents a supplementation to their diet. Barley provides a higher dry matter content and low amount of crude fibre, which confers to this feed high digestible and metabolisable energy. Nevertheless, we have rephrased this sentence, now it reads: “Therefore, a higher protein intake (provided by the barley grain supplementation) in fallow deer may be associated with increased levels or activity of some mitochondrial enzymes related to the sperm metabolism that could promote an enlargement of the sperm midpiece” (lines 339-341).

9. Line 498: At present, Table 1 only seems to be provided as background rather than as a supporting piece of evidence. This table is only cited once (within the Methods) and only protein and perhaps fat are alluded to. It is not yet clear why other nutrient analyses in the table were conducted and if these biochemicals might also contribute to the observed differences in the response variables. The arguments for higher-energy diet contributing to increased spermatogenic function and sperm midpiece length would also be strengthened if calorimetry findings or standard energy conversion estimates were to be included for the diets (e.g. protein=17 kJ/g, and fat=37 kJ/g; Raubenheimer and Rothman. 2013. Nutritional ecology of entomophagy in humans and other primates. *Annu Rev Entomol*).

On the basis of the estimated energetic values shown in Table 2, the findings have been rediscussed (lines 255-265).

Reviewer 2

We thank Reviewer 2 for the time taken to review our manuscript. We really appreciate his/her comments and suggestion, which helped us to improve our manuscript. Line numbers, where changes were, refer to the clean version of the manuscript.

Reviewer: 2

Comments to the Author(s)

In this work, the authors elicit the influence of an energy-rich diet on the onset of sexual maturation of fallow deer (*Dama dama*). To achieve this, the authors randomly allocated 32 pre-pubertal fallow deer to two groups: one fed with pasture and one supplemented with barley. Six months later, the animals were sacrificed and biometric parameters, serum testosterone, sperm parameters, spermatogenic function and acrosomal status were assessed. Overall, the authors report improvements in sperm parameters and reproductive function, compatible to an earlier onset of sexual maturation in the group fed with pasture supplemented with barley.

This paper presents some results that explore the potential commercial viability of fallow deer, based on an adequate methodology. The evaluated parameters are significant according to the aims of the work, and variables are properly defined. Besides, considering the amount of collected data and its further treatment, the reported results seem reliable.

Nevertheless, there are some major concerns regarding the way the authors present their data, and some gaps that must be filled in order to draw more consistent conclusions.

1. The authors have not identified the most critical aim of their work. The most evident conclusion of this work is that a high-energy diet anticipates the puberty onset on fallow deer. The authors do not present any conclusive proof that it was diet, and not sexual development during puberty, the responsible for the improvement in sperm parameters. In fact, even the bibliography cited by the authors raise the idea that it was the maturation state, likely due to nutritional state, the main differentiating factor between the two groups of deer. Therefore, the authors must rewrite the paper around this aim, which will forcibly mean changes in all sections, including title.

We would like to highlight that in our study fallow deer have the same age, so that the diet was the only difference between groups. Therefore, our results provide clear proofs that barley supplementation increases the testis mass, spermatogenic activity, sperm concentration, the percentage of the normal spermatozoa, and sperm midpiece length. As a plausible explanation, we suggest that high-energy diet may have fastened the onset of puberty. Nevertheless, as we state in the discussion (lines 304-307), in order to state that the dietary supplementation fasten the onset of puberty, repeated sample collections are required from the same individual throughout the breeding season.

2. The nomenclature of scientific terms is incoherent throughout the paper, rendering it unclear. As example, the nomenclature used for germ cell development phases is not coherent between Figure 1, Table 2 and the manuscript (the authors use round spermatocytes in text, then in table 2 they present counts for primary and secondary spermatocytes). Additionally, the authors should clearly define what are “sperm parameters”, “spermatogenic function” and “acrosomal status”.

We checked the manuscript and the term used were round spermatids and primary spermatocytes. We could not find “round spermatocytes”. As suggested by Reviewer 2, we clearly defined in the text the term “sperm parameters” (line 25 and lines 29-30) and we replaced the term “acrosomal status” with “acrosome integrity” (Lines 165, 214, 219) and the “spermatogenic function” with “spermatogenic activity” (line 24). However, we prefer to retain the term “spermatogenic function” in the title for clarity and conciseness as it is not easy to sum up all the results concerning the assessment of spermatogenesis in a brief statement.

3. Figure 1 is an essential figure in the context of the work but needs further refinement. It should show a wider field, or the authors could supplement it with an image of lower magnification, in order to show the seminiferous tubule structure and integrity.

Thank you for your suggestion. It is not possible to show the structure and integrity of seminiferous tubules because in our study the testicular samples were collected by fine needle aspiration cytology.

4. During discussion, the authors overestimate their observations based upon inadequate bibliography. It is not comparable sperm mid-piece size of lizards with deer, particularly if no measurement of mitochondria number and fitness was performed. Besides, it is not acceptable to relate that eventual gain to a diet effect, based upon data drawn from a study in rat liver. If the authors want to make this connection, they must present data on mitochondrial number and function in spermatozoa's midpiece.

Thank you for giving us the opportunity to better clarify this point. In the discussion section, we aim to provide a plausible explanation of the phenomenon observed in our study and not to compare species. To date, the relationship between nutrition and changes in the size of sperm structures is almost an unexplored topic. As a consequence, the references are extremely scarce. We believe therefore that the references cited in the manuscript are essential for the discussion of our findings. Moreover, all of them deal with vertebrate animals.

5. Since the authors collected serum of the animals, they should quantify other hormones of the reproductive axis, in order to further characterize the maturation phase of the deer at sacrifice.

Thank you for your suggestion. In this study, we evaluated the plasma testosterone levels, given the major role of this sexual hormone in male reproduction. Nevertheless, we have included in the discussion that: “Despite our study was focused on the effects of high-energy diet on blood testosterone levels, the role of other hormones like the follicle-stimulating hormone (FSH), thyroid hormone, insulin, and ghrelin should be considered in future studies, given their relevance in spermatogenesis and Sertoli cell functions (Oliveira and Alves 2015)” (lines 313-315).

6. Although the study focuses on fallow deer, the authors must not disregard works published in other mammal species, notably rodents and humans, where it is stated that high-energy diets lead to impaired sperm parameters and poorer fertility outcomes.

Thank you for your suggestions. We include in the manuscript references about the negative effect of obesity and undernourishment in male fertility in other species (rats: Jia et al. 2018; rams: Guan et al. 2014) (Lines 292-294).

7. I would not draw conclusions out of non-significant results as if there is a difference (e.g. testosterone levels).

We have not included in our conclusions the results about testosterone levels. We only state throughout the discussion that there was a tendency for higher blood testosterone levels in animals fed with barley grain than those fed with pasture only.

Additionally, there are minor issues that should be addressed when rewriting the article.

1. Please, review the superscripts in the boxplots and state what “a” and “b” stand for.

We rephrased the legends of Figure 3 and Figure 4 with “Different letters above the boxplots indicate significant differences ($p < 0.05$)”.

2. I advise the authors to look at the following bibliography to improve their work, especially regarding germ cell – Sertoli cell cooperation: PMID 23495257, 29453570; ISBN 978-3-319-19791-3.

Thank you for your suggestion. All references have been included in the manuscript (lines 274 and 315).

Appendix B

Response to Referees

Dear Editors and Reviewers, we would like to thank you again for the time taken to review our manuscript. We really appreciate your comments and suggestion, which greatly helped us to improve our manuscript. As suggested by Reviewer 2, we included in the title of our manuscript the term “yearlings”, which refers to fallow deer that are 1 to 2 years old. The term is also included in the conclusions. Moreover, we specified that superscripts indicate significant differences between groups in Figures 3 and 4.

Further changes are detailed below (lines refer to the clean version of the manuscript):

Lines 19-20 We included Dr. Ros Santaella as author for correspondence

Line 33 We added the term “sperm” to “midpiece”

Line 105 We deleted the comma

Line 123 We added the term “hours”

Lines 173-174 We replaced the term “phenotype” with “morphology”, for consistency with the Figure 2. We also deleted the term “normal” and replaced the term “assessed” with “evaluated”

Line 183 We corrected the typo in “Shapiro-Wilk”. We also delete the dash from “paired sample”

Line 216 We added “for control and high-energy diet groups” to the sentence

Line 255 We added to the sentence “using the FNAC technique”

Line 273 We added “cell” to “Sertoli index”

Line 283 We replaced “it remains to be tested the effects of high-energy diet in testicular metabolism” with: “it remains to determine the effects of high-energy diet on the testicular metabolism”

Lines 288-291 “Sertoli cell” instead of “Sertoli cells”. We replaced “their” with “its”.

Line 329-330 We replaced “between” with “among” and added the dot to “vs”

Line 343 We included “involved in” to the sentence

Line 356 We replaced “reproductive function” with “reproductive performance”